# An effective protocol to isolate and mechanically test silk fibers spun by *Osmia lignaria* Say (Hymenoptera: Megachilidae) fifth instar larvae

Oran Wasserman[1◉], Jackson J. Morley[2◉], Mary-Kate F. Williams[1,3◉], Brianne E. Bell[1], Diana L. Cox-Foster[3*], Justin A. Jones[1*]

1 Department of Biology, Utah State University, Logan, Utah, USA, 2 Department of Chemistry and Biochemistry, Utah State University, Logan, Utah, USA, 3 Pollinating Insect Research Unit, Pacific West Area, Agricultural Research Service, United States Department of Agriculture, Logan, Utah, USA

◉ Authors contributed equally
* Diana.Cox-Foster@usda.gov (DLCF); Justin.A.Jones@usu.edu (JAJ)

## Abstract

Silk, a remarkable protein-based fiber spun by various arthropod lineages, has been prized for millennia, with the cocoon silk of domesticated silkworms and spiders being the most utilized and extensively studied. There is limited information on how silk can be used to investigate biology, development, and health in other silk-producing species, particularly for solitary bees such as *Osmia lignaria* Say (Hymenoptera: Megachilidae). *Osmia lignaria*, an increasingly managed solitary pollinator, produces silk cocoons during the fifth instar larval stage. We have developed a minimally invasive protocol to isolate and mechanically test *O. lignaria* silk fibers using a 3-D printed well plate system for rearing and two specific isolation techniques. Our protocol allows for collecting individual fibers directly from silk-spinning larvae between silk initiation and cocoon formation without preventing subsequent cocoon development, enabling silk characterization as part of larger rearing and developmental studies. For this study, isolated fibers were mounted on C-cards, facilitating diameter measurement using a microscope and mechanical testing with an MTS Synergie 100 tensile testing instrument. We successfully isolated and tested the mechanical properties of naturally spun silk from *O. lignaria*, with 20 fibers isolated and mechanically tested from seven larvae. Further examination of isolated silk can reveal physical, molecular, chemical, and morphological characteristics, advancing our understanding of bee silk properties and their role in bee biology, evolution, and nutritional status. This protocol provides a practical tool for researchers to isolate and study silk from silk-producing bee species.

## Introduction

For thousands of years, humans have incorporated silk into textile production, cultural heritage items, and medical applications [1–2]. Different insect orders produce silk for nests, protection, shelter, and other functions [3–5]. Insect silk varies in size, amino acid composition,

**Data availability statement:** All relevant data
are within the manuscript and its Supporting
Information files.

**Funding:** This work was supported by scholar-
ships from the Department of Biology at Utah
State University and a grant from the Utah
State University Office of Research. Funds from
the U.S. Department of Agriculture, Agricultural
Research Service (Project#2080-21000-019-
000-D) supported rearing the O. lignaria larvae.
The funders had no role in study design, data
collection and analysis, decision to publish, or
preparation of the manuscript.

**Competing interests:** The authors have
declared that no competing interests exist.

and production varies across different life stages and species [5]. Silk has been used as an indicator for silkworms (*Bombyx mori* L. [Lepidoptera: Bombycidae]) and spiders (Arachnida: Araneae Clerck) to determine the nutritional status [6–8] and pesticide exposure [9–10]. However, similar studies on other silk-producing insects, such as bees, are limited.

*Osmia lignaria* Say (Hymenoptera: Megachilidae), commonly referred to as the blue orchard bee, is a native solitary bee species in North America that plays an important role in orchard pollination [11]. Understanding the biology of *O. lignaria* is essential for the development of management strategies to support their populations for pollination efforts [12–15]. One aspect of *O. lignaria* biology that has received limited attention is the production and properties of their silk cocoons. *O. lignaria* produces silk cocoons during the late fifth larval instar stage [11]. By late spring, the fifth instar completes the consumption of a pollen-nectar provision, defecates, and spins silk strands extruded from the labial glands [16–17]. Due to air exposure, the extruded fibers harden from a silk solution to a solid silk fiber [1,18–19]. This is repeated until a thick mesh of threads forms the cocoon.

By enabling comparative analyses of silk properties across different bee lineages, this method can shed light on the evolutionary and ecological factors shaping silk production and on silk properties such as covalent crosslinking in bees [20]. Moreover, pollinators, including bees, play a crucial role in food security, economic stability, maintaining ecosystems, and supporting agriculture productivity [21–24]. However, many pollinator populations are declining due to multiple stressors [25]. Potential stressors such as pesticides, nutrition requirements, and others can potentially be detected using silk as an indicator, as was done for silkworms and spiders.

Current published bee silk protocols have reported different methodologies for isolating and mechanically testing bee silk, such as transferring larvae to glass slides inside a box [26] or cutting sections from individual honeybee cocoon cell walls, which required extra processing to remove bee wax [27]. A detailed explanation and demonstration of an effective isolation method that does not require large specimen numbers or extra processing post-silk isolation is missing. Our method introduces a novel approach to isolating *O. lignaria* silk fibers by utilizing a 3-D printed well plate system for rearing larvae and two specific isolation techniques. This method allows for the collection of individual fibers directly from the silk-spinning larvae, minimizing larval cocoon-formation interference and the need for large specimen pools or extra post-processing steps.

The objective of this study is to present a detailed protocol for isolating and mechanically testing silk fibers produced by *O. lignaria*. By providing a step-by-step guide for fiber isolation, diameter measurement, and mechanical testing, we aim to establish a standardized method that can be adapted for studying silk from other bee species. Our protocol offers a minimally invasive approach that allows larvae to proceed with cocoon formation after fiber isolation, making it suitable for integration into larger rearing experiments. This protocol advances our understanding of *O. lignaria* silk properties and provides a framework for investigating silk from other understudied bee species. We hope that by using this protocol, comparative analyses of silk properties across different bee lineages can shed light on evolutionary and ecological factors shaping silk use and production in bees. The ability to isolate and characterize bee silk fibers without disrupting the larvae's natural cocoon formation process opens up new avenues for investigating the relationship between silk biochemistry and mechanical properties.

## Materials and methods

No approval of research ethics committees was required to accomplish the goals of this study because experimental work was conducted with an unregulated invertebrate species.

The protocol described in this peer-reviewed article is published on protocols.io (https://doi.org/10.17504/protocols.io.bp2l6d8bkvqe/v1) and is included for printing purposes as S1-S4 Files. Material manufacturers and item or reference numbers are listed in the S5 File.

## Bee development

### Materials

- 3-D printed well plate(s)

- Dissection microscope

- Excised *O. lignaria* nest cells

The well plate rearing system utilized in this experiment was modified from a protocol written by Williams et al. (2024) [28]. In an apple grove in Logan, Utah, USA, *O. lignaria* were trapped using artificial nest materials, bundles of cardboard tubes with straw inserts hung in a corrugated plastic nesting box [29]. In June 2023, completed nests were transferred to a laboratory, and developing bees inside sealed nest cells were excised from the straw. Nest cells contained second instars, a provision (food source of nectar and pollen provided by the mother) [20], and the original straw insert (Fig 1A). Excised nest cells were placed into a 3-D printed well plate to hold the developing bees and allow them to be visually monitored for development. Larval development was examined daily using a dissection microscope. Wells with developing bees were examined using 20X total magnification and with increased magnification to peer deeper into the well when necessary. Once fecal pellets were detected, indicating a fifth instar larva, signs of cocoon initiation were monitored more frequently. Cocoon initiation begins when late fifth instar larvae make circular motions to produce silk from salivary glands [30] and can be observed less than 24 hours after the provision is mostly consumed [31–32]. If larvae exhibited abnormal behavior, discoloration, or pigmentation, they were removed from the well plate.

## Fiber isolation and mounting onto C-cards

### Materials

- Black glass mat

- Dissection forceps

- Dissection microscope with an external light source

- Dissection scissors

- Microscope slide

- Single-edge razor blade

- Super glue

- Transparent tape

- Wood toothpick 10.16 centimeters (cm)

- X-ray film sheets C-cards (Fig 1E)

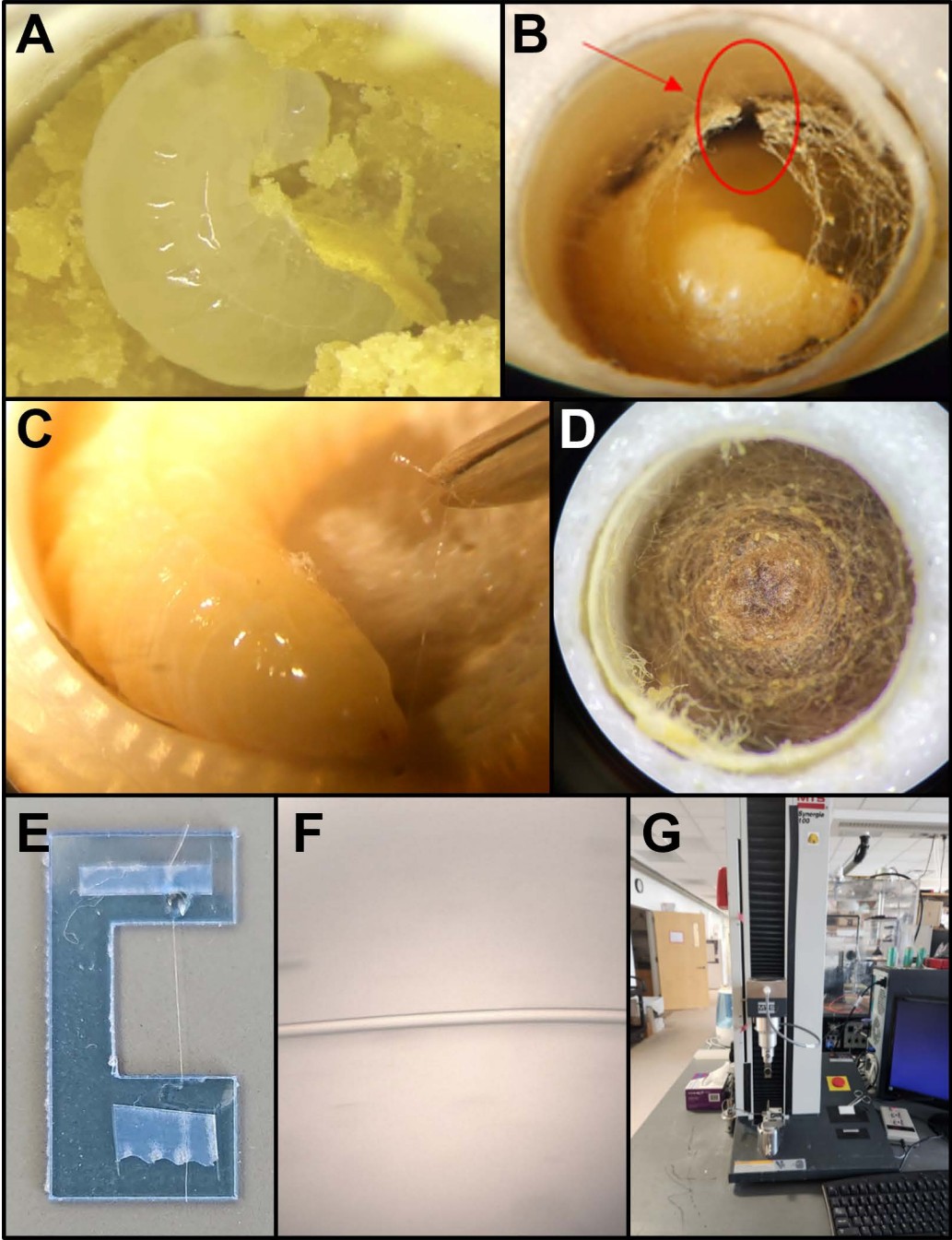

**Fig 1. Overview of significant *O. lignaria* life stages and protocol steps for this study.** (A) and (B) are images of the two life stages used in this study: the second instar and the late fifth instar beginning to spin silk, respectively. The red arrow and circle indicate a cut made in the silk mesh to access the larva's mouth. **(C)** A single silk strand is isolated from the larva by pulling silk directly from the larva's mouth using dissecting forceps. (D) Completed *O. lignaria* cocoon (top-down view inside of rearing well plate). (E) isolated and carded fiber onto a plastic C-card. (F) Isolated and carded fiber under a light microscope at 400X total magnification. (G) MTS Synergie 100 tensile testing instrument. (A) and (D) are images taken by Williams et al. (2024) [20] and replicated with permission of the authors.

Wells were positioned under a dissection microscope starting at 20X total magnification and increasing the magnification to look deeper into the well to observe silk strands produced at the bottom of the well once the late fifth instar started to extrude silk [31]. The first few hours of silk spinning are often intermittent; during this time, the larva seems most permissive of manual fiber isolation. This optimal timing is evident in a circular mesh of silk that begins to appear in the well, as seen in Fig 1B. Once the circular mesh of silk appeared, using dissecting scissors, appropriate cuts were made in the mesh (see the red arrow in Fig 1B as an example of the cut), and the flaps were folded back (Fig 1B). In some cases, excess silk was removed using forceps, which gave better access to the larva.

The well containing the silk-spinning larva was positioned under a dissecting microscope with 20X total magnification. Two isolation techniques were utilized to grasp the silk with forceps. The first isolation technique was to use forceps to pinch the lengthening fiber as the larva pulled the fiber from one wall connection to another, as seen in Fig 1C. The second isolation technique involved positioning the forceps in front of the larva or gently touching the forceps to the mouth or head of the larva, prompting the attachment of newly spun silk strands to the forceps. Once the fiber was pinched, the fiber was gently pulled at speed similar to the speed at which the larva moved its head to the other wall, directing the larva to pull away from the forceps instead of from the straw. Pulling the fiber at the same speed as the larvae are extruding the fiber minimizes unnecessary stretching to conserve the natural fiber qualities. On average, a fiber of approximately 15 cm in length was collected before the fiber strand broke. This provides a single fiber for mounting onto plastic C-shaped sample cards for mechanical testing or for further fiber analysis and testing. This is visualized in the S1 Video.

The isolated fiber was transferred to a black glass mat. Clear tape (around 2 millimeters [mm] by 5 mm) held with forceps was used to apply tape perpendicularly to both ends of the fiber. Once an isolated fiber was taped down, a roughly sized 4 mm x 5 mm piece of tape was applied around 12 mm away from the end tape so the fiber would span the C-card gap. While still on the glass mat, a razor blade was used to cut down the middle of the widest portion of the tape, separating a shorter fiber segment from the isolated fiber. One end of a C-card was positioned alongside the shorter fiber between the two pieces of tape. Then, one piece of tape/fiber was placed on one end of the C-card using a pair of forceps. The same process was repeated for the other end of the fiber before securing the C-card to a microscope slide with tape for safe transfer. Using a toothpick, super glue was added on top of the isolated fiber between the tape and the C-card gap to secure it onto the C-card [33–34]. The glue was allowed to dry for at least 30 minutes before mechanical testing. The above steps were repeated until a total of 25 fiber sections were carded. S2 Video and S3 Video display these processes.

### Fiber diameter measurement and mechanical testing

#### Materials

- Dissection scissors

- Microscope slide

- Motic BA310 microscope and the Motic Image Plus 3.1 program (Schertz, TX, USA)

- Single-edge razor blade

- Tensile testing equipment (Material Testing System Synergie 100 tensile testing instrument with a custom 10 grams (g) load cell [Transducer Techniques, Newark, CA, USA])

Each carded fiber (Fig 1E) was placed under a Motic BA310 microscope with a 400x total magnification to measure the fiber diameter. Using the Motic Image Plus 3.1 program, three diameter measurements were taken at three different locations on the fiber (both ends and the middle) to account for variations in natural fiber diameter, resulting in a total of nine measurements per fiber. The nine total measurements were averaged and recorded. After recording the diameter, each carded fiber was removed from the microscope slide by cutting the tape with a razor blade. The C-card was loaded and tested in an MTS Synergie 100 tensile testing instrument. Loading was done by inserting the C-card into the clamps, lowering the top clamp, causing the C-card to bow and fiber to go slack, cutting the C-card with scissors, and returning the clamps to a height with the fiber nearly taut. The raw data was exported (S7 File) and analyzed in Microsoft Excel to calculate each fiber's maximum tensile stress, maximum strain, elastic modulus, and toughness (S6 File).

## Results

Using the protocol above, we successfully isolated and tested the mechanical properties of naturally spun silk from *O. lignaria*. Overall, 25 fibers were isolated and placed on cards; of these, five fibers broke in the testing process, resulting in 20 fibers that were tested from seven larvae (Table 1, Table S7–1). The larvae in this study proceeded to cocoon formation, demonstrating that our method is minimally invasive and does not hinder natural development. During the isolation process, we observed that *O. lignaria* larvae attach silk strands to the cell wall and then pull away with random head movements until the larvae attach the silk to another part of the cell wall. The larva will then touch another part of the wall, cementing the fiber in the new location. *Osmia lignaria* larvae spun silk in a similar behavior and process to honeybee larvae, as described by Jay (1964) [41]. We suspect how the larvae spin the fibers results in un-uniform fibers that vary in diameter. The largest diameter observed was 46.4 micrometers (μm), and the smallest was 1.9 μm. This large variation in diameter was observed in other natural fibers [35]. The variation in diameter translates to the large error bars observed in the mechanical properties since the diameter is included in the equations to calculate mechanical properties.

## Discussion

Our initial attempts to isolate and characterize *O. lignaria* silk fibers were conducted using fully formed cocoons. However, this approach proved to be laborious and yielded inconsistent results due to the hardening of the silk onto surfaces or adjacent fibers [19]. The silk fibers in a completed cocoon are tightly packed and fused [36], and silk becomes brittle with increased exposure to the environment [37]. These challenges prompted us to develop a more efficient and reliable method for isolating *O. lignaria* silk fibers in their natural state. By directly accessing the silk-spinning larvae during the early stages of cocoon formation, we were able to collect individual fibers before they fused with other fibers in a formed cocoon [36]. This approach significantly improved the ease and consistency of fiber isolation, allowing for a more accurate characterization of the properties of *O. lignaria* silk.

Studying bee silk can provide insight into their biology. Isolated silk can be analyzed via proteomics tools [38–40] to determine divergence and protein structure or to determine the

Table 1. Mechanical property results of isolated silk fibers from *O. lignaria* larvae (n = 7), in which we received 20 fibers to test. Measurements are reported means ± standard deviation.

| Diameter (μm) | Toughness (MJ m$^{-3}$) | Tensile strength (MPa) | Strain (mm mm$^{-1}$) | Elastic modulus (GPa) |
|---|---|---|---|---|
| 6.87 ± 2.22 | 6.18 ± 4.99 | 66.43 ± 35.07 | 0.17 ± 0.11 | 4.84 ± 4.11 |

presence of eDNA to assess biodiversity in different environments [41]. Honeybee silk proteins have been identified and characterized, providing a foundation for further research into the structure and function of these proteins compared to other bee silk proteins [42]. Comparing the silk of solitary and social bees may reveal evolutionary adaptations and the underlying genetic mechanisms. For example, social bees like honey bees and bumble bees live in colonies and produce silk for different purposes, such as constructing brood cells and protecting the hive [43]. In contrast, solitary bees like *O. lignaria* use silk primarily for constructing cocoons within nests [11]. Analyzing the mechanical properties, amino acid composition, phylogenetic variation, gene expression patterns, and biological plasticity of silk from these different bee species may provide insights into how silk production has evolved to suit specific life cycles and nesting behaviors, such as was demonstrated for spider silks [44]. Our protocol enables the isolation of silk fibers from individual larvae, allowing for such comparative analyses to be conducted.

There are limitations that can affect the effectiveness of our protocol when applied to species other than *O. lignaria*. The rearing and isolation methods reported in this protocol are tailored to *O. lignaria* and may require modification for other species. When applying our protocol to other bee species, first, a thorough investigation of the designated bee species should address their size, life cycle, and the manner in which they construct cocoons. In the rearing stage, the well diameters and potentially other specifications will need to be modified for bees that are physically smaller than *O. lignaria*, such as *Osmia bruneri* Cockerell, *Osmia ribifloris* Cockerell, and *M. rotundata*. Also, a smaller well diameter might make it challenging to access the larva's mouth with forceps or inflict an injury if mishandled, which will impede the silk isolation process and reduce the number of samples in an experiment.

During the mechanical testing stage, several factors can influence the mechanical properties of native natural fibers, such as mounting techniques, source and processing conditions, and testing environmental factors, such as moisture and temperature [35,45–50]. Future studies could investigate how these environmental factors specifically affect the *O. lignaria* silk spinning process and the resulting fiber properties.

Another point to consider is that native silk fibers have been historically reported to have a higher degree of variability in diameter and mechanical properties, which is not caused by the sample size [35]. Given the limitations regarding mechanical testing of native silk fibers, we recommend including comprehensive data regarding the fiber's isolation techniques and testing parameters to design future protocols and report data that can enable meaningful comparisons and conclusions between different species, fiber isolation techniques, testing parameters, and environmental conditions.

The challenges arising from the decrease in pollinator populations, such as pesticide usage, diseases, and habitat loss, require innovative approaches to identify and assess the factors affecting bee health [51]. Bee silk has been investigated for versatility in various biomedical applications [52] but not yet as a tool to assess bee health [26]. Overall, the procedure described in this paper can be used to isolate and test silk from a variety of solitary bee species whose silk has not been characterized yet, such as *Osmia chalybea* Smith, *Osmia californica* Cresson, *Osmia montana* Cresson, *Megachile rotundata* (F.) (Hymenoptera: Megachilidae), *Stelis ater* Mitchell (Hymenoptera: Megachilidae), and others [11,16,30,53]. However, the *Bee Development* and *Fiber Isolation* steps should be modified to account for differences among solitary bee species, especially smaller species where well diameters need to be adjusted, and silk acquisition may be more or less challenging compared to the methods presented for *O. lignaria*. By enabling the study of bee silk properties and potential applications, our protocol offers a new approach to addressing the challenges

faced by pollinator populations and supporting conservation efforts. In addition, investigating the chemical and mechanical properties of solitary bee silk can shed light on the function of the cocoons in protection against parasitoids and protection from the environment [54].

## Conclusion

In summary, our study presents a novel, minimally invasive protocol for isolating and mechanically testing silk fibers from the solitary bee species *O. lignaria* that will also work for other bee species. This protocol enables the collection of individual fibers directly from silk-spinning larvae, allowing for the characterization of bee silk properties without disrupting the natural cocoon formation process. By providing a practical tool for investigating bee silk, our work opens up new avenues for studying the role of silk in bee biology and evolution. Furthermore, the insights gained from understanding bee silk properties may contribute to the conservation efforts of these essential pollinators. By applying our method to a diverse range of silk-producing bee species, researchers can unravel the complexities of bee silk and its potential applications both as a biomaterial and in understanding how the cocoons protect the developing bees.

## Supporting information

**S1 File. Step-biky-step protocol collection.** https://doi.org/10.17504/protocols.io.bp2l6d8bkvqe/v1
(PDF)

**S2 File. Step-by-step protocol for C-cards preparations.** https://doi.org/10.17504/protocols.io.ewov1d28pvr2/v1
(PDF)

**S3 File. Step-by-step protocol for Fiber Isolation and Mounting onto C-cards.** https://doi.org/10.17504/protocols.io.x54v9r7mqv3e/v1
(PDF)

**S4 File. Step-by-step protocol for Fiber Diameter Measurement and Mechanical Testing.** https://doi.org/10.17504/protocols.io.eq2ly6x3mgx9/v1
(PDF)

**S5 File. Detailed materials list.**
(XLSX)

**S6 File. Guide for mechanical properties analysis.**
(DOCX)

**S7 File. Mechanical properties of individual isolated silk fibers and MTS raw data.**
(XLSX)

**S1 Video. Fiber isolation.**
(MP4)

**S2 Video. C-card fiber mounting.**
(MP4)

**S3 Video. Fiber mounting using a dissection microscope.**
(MP4)

## Acknowledgments

Thank T. Pitts-Singer and the Behavior Lab at USDA-ARS-PWA Pollinating Insect Research Unit (PIRU) for their help setting up nesting materials and providing populations of *O. lignaria* for research. We also thank PIRU for using their apple stand to acquire nests.

## Author contributions

**Conceptualization:** Oran Wasserman, Jackson J. Morley, Mary-Kate F. Williams, Justin A. Jones.

**Formal analysis:** Oran Wasserman, Jackson J. Morley, Brianne E. Bell.

**Funding acquisition:** Oran Wasserman, Jackson J. Morley, Diana L. Cox-Foster.

**Investigation:** Oran Wasserman, Jackson J. Morley, Mary-Kate F. Williams.

**Methodology:** Oran Wasserman, Jackson J. Morley, Mary-Kate F. Williams.

**Project administration:** Justin A. Jones.

**Resources:** Diana L. Cox-Foster, Justin A. Jones.

**Supervision:** Justin A. Jones.

**Writing – original draft:** Oran Wasserman, Jackson J. Morley, Mary-Kate F. Williams.

**Writing – review & editing:** Oran Wasserman, Jackson J. Morley, Mary-Kate F. Williams, Brianne E. Bell, Diana L. Cox-Foster, Justin A. Jones.

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
