## [Decision Letter · Decision Letter 0]

26 Nov 2024

PONE-D-24-25883An effective protocol to isolate and mechanically test silk fibers spun by Osmia lignaria Say (Hymenoptera: Megachilidae) fifth instar larvaePLOS ONE

Dear Dr. Wasserman,

Thank you for submitting your manuscript to PLOS ONE. After careful consideration, we feel that it has merit but does not fully meet PLOS ONE’s publication criteria as it currently stands. Therefore, we invite you to submit a revised version of the manuscript that addresses the points raised during the review process.

**ACADEMIC EDITOR: **The study initiated by Wasserman et al. is well elaborated and written. Some minor points as raised by the reviewer need to be addressed before it can be made available to the general audience.

I thank you once again for your patience and I am looking forward to received your newer version of the manuscript incorporating the suggested improvements.

We look forward to receiving your revised manuscript.

Kind regards,

Adel Tekari, PhD

Academic Editor

PLOS ONE

2. We note you have not yet provided a protocols.io PDF version of your protocol and/or a protocols.io DOI. When you submit your revision, please provide a PDF version of your protocol as generated by protocols.io (the file will have the protocols.io logo in the upper right corner of the first page) as a Supporting Information file. The filename should be S1_file.pdf, and you should enter “S1 File” into the Description field. Any additional protocols should be numbered S2, S3, and so on. Please also follow the instructions for Supporting Information captions [https://journals.plos.org/plosone/s/supporting-information#loc-captions]. The title in the caption should read: “Step-by-step protocol, also available on protocols.io.”

Please assign your protocol a protocols.io DOI, if you have not already done so, and include the following line in the Materials and Methods section of your manuscript: “The protocol described in this peer-reviewed article is published on protocols.io (https://dx.doi.org/10.17504/protocols.io.[...]) and is included for printing purposes as S1 File.” You should also supply the DOI in the Protocols.io DOI field of the submission form when you submit your revision.

If you have not yet uploaded your protocol to protocols.io, you are invited to use the platform’s protocol entry service [https://www.protocols.io/we-enter-protocols] for doing so, at no charge. Through this service, the team at protocols.io will enter your protocol for you and format it in a way that takes advantage of the platform’s features. When submitting your protocol to the protocol entry service please include the customer code PLOS2022 in the Note field and indicate that your protocol is associated with a PLOS ONE Lab Protocol Submission. You should also include the title and manuscript number of your PLOS ONE submission.

“The Department of Biology at Utah State University and the Utah State University Office of Research funded and supported this work. Funds from the U.S. Department of Agriculture, Agricultural Research Service (Project#2080-21000-019-000-D) supported the rearing of the O. lignaria larvae.”

Additional Editor Comments:

None

Reviewers' comments:

Reviewer's Responses to Questions

**Comments to the Author**

1. Does the manuscript report a protocol which is of utility to the research community and adds value to the published literature?

Reviewer #1: Yes

Reviewer #2: Yes

2. Has the protocol been described in sufficient detail?

To answer this question, please click the link to protocols.io in the Materials and Methods section of the manuscript (if a link has been provided) or consult the step-by-step protocol in the Supporting Information files.

The step-by-step protocol should contain sufficient detail for another researcher to be able to reproduce all experiments and analyses.

Reviewer #1: Yes

Reviewer #2: Yes

3. Does the protocol describe a validated method?

Reviewer #1: Yes

Reviewer #2: Yes

4. If the manuscript contains new data, have the authors made this data fully available?

Reviewer #1: Yes

Reviewer #2: Yes

**5. Is the article presented in an intelligible fashion and written in standard English?**

Reviewer #1: Yes

Reviewer #2: Yes

6. Review Comments to the Author

Reviewer #1: The protocol for isolation and mechanical testing of silk fibers spun by the blue orchard bee, an important pollinator of orchards in North America, offers a minimally invasive approach that allows larvae to proceed with cocoon formation after fiber isolation. This protocol allows larval rearing of this ecological important insects and thus, it is very important for investigating the relationship between silk biochemistry and mechanical properties without killing the bees. In addition, the presented protocol can also be used for testing of silk from other solitary bee species. Therefore, the relevance of this protocol is very high. The protocol is very detailed and easy to follow because of the very detailed description of the single steps. In addition, the discussion of possible limitations is very helpful to avoid errors in the implementation of the protocol by other scientists at an early stage.

Reviewer #2: The authors needs to answer certain queries provided which can be included in the manuscript:

1. What are the alternative materials that can replace 3-D printed well plates for rearing Osmia lignaria larvae?

2. How does the timing of fiber isolation affect the mechanical properties of the silk?

3. What are the optimal environmental conditions (temperature, humidity) for silk spinning in Osmia lignaria?

4. How can the isolation method be adapted for smaller bee species with different cocoon construction methods?

5. Does the method of cutting the silk mesh impact the tensile strength or elasticity of the isolated fibers?

Mechanical Testing

6. How do variations in fiber diameter correlate with mechanical properties like tensile strength and elasticity?

7. Can the tensile testing protocol be adapted for higher-throughput analyses of multiple fibers simultaneously?

8. What are the effects of different glues on the mechanical properties of silk during C-card mounting?

9. How does the choice of testing environment (e.g., ambient vs. controlled humidity) affect mechanical test outcomes?

Silk Characteristics and Applications

10. What molecular markers in the silk indicate the health and nutritional status of the larvae?

11. How does the chemical composition of Osmia lignaria silk compare to that of other bee or silkworm species?

12. Can the silk from Osmia lignaria be used as a biomarker for exposure to environmental stressors, such as pesticides?

13. What are the biodegradation properties of Osmia lignaria silk in various environmental conditions?

Comparative Studies

14. How do the mechanical properties of silk from solitary bees compare with those of social bees, such as honeybees?

15. What are the evolutionary implications of the observed mechanical differences between Osmia lignaria silk and other insect silks?

16. Are there phylogenetic correlations between the mechanical properties of bee silks and their nesting behaviors?

Protocol Optimization

17. What improvements can be made to reduce the variability in fiber diameter during isolation?

18. How does the larval diet during development influence the quality and mechanical properties of the silk?

19. Can the protocol be modified to allow for automated fiber isolation?

Broader Applications

20. What are the potential industrial applications of silk isolated using this protocol (e.g., biomedical materials, eco-friendly textiles)?

7. PLOS authors have the option to publish the peer review history of their article (what does this mean? ). If published, this will include your full peer review and any attached files.

**Do you want your identity to be public for this peer review?** For information about this choice, including consent withdrawal, please see our Privacy Policy .

Reviewer #1: No

Reviewer #2: No

---

## [Author Response · Author response to Decision Letter 1]

6 Jan 2025

We appreciate the care and time the reviewers have taken in their comments on our manuscript. We have provided a detailed response to all the comments assigned to the manuscript. If additional points in the manuscript need to be revised, please let us know, and we will be happy to provide an explanation or a revision.

---

## [Decision Letter · Decision Letter 1]

24 Jan 2025

An effective protocol to isolate and mechanically test silk fibers spun by Osmia lignaria Say (Hymenoptera: Megachilidae) fifth instar larvae

PONE-D-24-25883R1

Dear Dr. Wasserman,

We’re pleased to inform you that your manuscript has been judged scientifically suitable for publication and will be formally accepted for publication once it meets all outstanding technical requirements.

Kind regards,

Adel Tekari, PhD

Academic Editor

PLOS ONE

Additional Editor Comments (optional):

Reviewers' comments:

Reviewer's Responses to Questions

**Comments to the Author**

1. Does the manuscript report a protocol which is of utility to the research community and adds value to the published literature?

Reviewer #1: Yes

Reviewer #2: Yes

2. Has the protocol been described in sufficient detail?

To answer this question, please click the link to protocols.io in the Materials and Methods section of the manuscript (if a link has been provided) or consult the step-by-step protocol in the Supporting Information files.

The step-by-step protocol should contain sufficient detail for another researcher to be able to reproduce all experiments and analyses.

Reviewer #1: Yes

Reviewer #2: Yes

3. Does the protocol describe a validated method?

Reviewer #1: Yes

Reviewer #2: Yes

4. If the manuscript contains new data, have the authors made this data fully available?

Reviewer #1: Yes

Reviewer #2: Yes

**5. Is the article presented in an intelligible fashion and written in standard English?**

Reviewer #1: Yes

Reviewer #2: Yes

6. Review Comments to the Author

Reviewer #1: There is nothing to add to my original review. Thus I recommend to accept the manuscript in the revised version.

Reviewer #2: 1. What statistical methods were employed to analyze the large variation in mechanical property data presented in Table 1, and how were the error margins validated?

2. What measures were taken to ensure the reproducibility of fiber isolation and testing techniques across different trials or by different researchers?

3. How do the mechanical properties of Osmia lignaria silk fibers compare to those of other insect silk (e.g., silkworms or spiders) as discussed in the results and supported by the data in Table 1?

7. PLOS authors have the option to publish the peer review history of their article (what does this mean? ). If published, this will include your full peer review and any attached files.

**Do you want your identity to be public for this peer review?** For information about this choice, including consent withdrawal, please see our Privacy Policy .

Reviewer #1: No

Reviewer #2: No

---

## [Editor Report · Acceptance letter]

PONE-D-24-25883R1

PLOS ONE

Dear Dr. Wasserman,

I'm pleased to inform you that your manuscript has been deemed suitable for publication in PLOS ONE. Congratulations! Your manuscript is now being handed over to our production team.

Kind regards,

on behalf of

Dr. Adel Tekari

Academic Editor

PLOS ONE